# Integrated Care as a Model for Interprofessional Disease Management and the Benefits for People Living with HIV/AIDS

**DOI:** 10.3390/ijerph20043374

**Published:** 2023-02-15

**Authors:** Helmut Beichler, Igor Grabovac, Thomas E. Dorner

**Affiliations:** 1Nursing School, Vienna General Hospital, Medical University of Vienna, 1090 Vienna, Austria; 2Department of Social and Preventive Medicine, Centre for Public Health, Medical University of Vienna, 1090 Vienna, Austria; 3Academy for Ageing Research, Haus der Barmherzigkeit, 1090 Vienna, Austria

**Keywords:** HIV/AIDS, integrated care, chronic diseases, adherence, disease management, case management, stigmatizing, discrimination, complication, cognitive disorder

## Abstract

Introduction: Today, antiretroviral therapy (ART) is effectively used as a lifelong therapy to treat people living with HIV (PLWH) to suppress viral replication. Moreover, PLWH need an adequate care strategy in an interprofessional, networked setting of health care professionals from different disciplines. HIV/AIDS poses challenges to both patients and health care professionals within the framework of care due to frequent visits to physicians, avoidable hospitalizations, comorbidities, complications, and the resulting polypharmacy. The concepts of integrated care (IC) represent sustainable approaches to solving the complex care situation of PLWH. Aims: This study aimed to describe the national and international models of integrated care and their benefits regarding PLWH as complex, chronically ill patients in the health care system. Methods: We conducted a narrative review of the current national and international innovative models and approaches to integrated care for people with HIV/AIDS. The literature search covered the period between March and November 2022 and was conducted in the databases Cinahl, Cochrane, and Pubmed. Quantitative and qualitative studies, meta-analyses, and reviews were included. Results: The main findings are the benefits of integrated care (IC) as an interconnected, guideline- and pathway-based multiprofessional, multidisciplinary, patient-centered treatment for PLWH with complex chronic HIV/AIDS. This includes the evidence-based continuity of care with decreased hospitalization, reductions in costly and burdensome duplicate testing, and the saving of overall health care costs. Furthermore, it includes motivation for adherence, the prevention of HIV transmission through unrestricted access to ART, the reduction and timely treatment of comorbidities, the reduction of multimorbidity and polypharmacy, palliative care, and the treatment of chronic pain. IC is initiated, implemented, and financed by health policy in the form of integrated health care, managed care, case and care management, primary care, and general practitioner-centered concepts for the care of PLWH. Integrated care was originally founded in the United States of America. The complexity of HIV/AIDS intensifies as the disease progresses. Conclusions: Integrated care focuses on the holistic view of PLWH, considering medical, nursing, psychosocial, and psychiatric needs, as well as the various interactions among them. A comprehensive expansion of integrated care in primary health care settings will not only relieve the burden on hospitals but also significantly improve the patient situation and the outcome of treatment.

## 1. Introduction

In the European Union (EU), approximately 25,000 people were newly diagnosed with HIV in 2021 [1,2]. A total of 85% of newly detected cases are diagnosed at a late disease stage (defined as less than 350 CD4 cells per mm^3^) [3]. Compared to low-income countries, people living with HIV (PLWH) in Europe mostly have unrestricted access to antiretroviral therapy (ART). This therapy has fundamentally changed the course of the HIV pandemic due to an effective lifelong viral replication suppression [4,5]. Furthermore, ART has significantly reduced mortality and morbidity [6,7,8]. ART also reduces opportunistic infections and the incidences of comorbidities, multimorbidity, and complications [9]. The requirement for this is to achieve adherence of at least 95% [10]. Despite the normalization of life expectancy, the incidences of carcinoma, cardiovascular disease, liver disease, kidney disease, diabetes mellitus, chronic pain with opiate abuse, and hypertension accumulate to a complex multifactorial situation [11,12,13,14,15]. Health care systems are currently characterized by many actors operating in different sub-areas. Thus, physicians in private practice, institutes, laboratories, and social insurance institutions, as well as in outpatient and inpatient areas of hospitals, are involved in prevention and diagnosis to therapy and rehabilitation. Patients therefore pass through a series of care transitions and interfaces within the health care system, which must be overcome. This is where the concept of integrated care comes in. The aim is to optimize transitions between different areas of treatment and specialties and to make the interfaces between the sub-areas holistic, efficient, effective, and patient-oriented.

Complex chronic diseases pose significant challenges to health care professionals in the management of PLWH in the 21st century and necessitate a redesign of health care systems [16]. Frequent physician visits, avoidable hospitalizations, polypharmacy, and an increasing need for long-term care are the consequences of an unstructured, cost-intensive treatment that is burdensome for patients and health care systems [17], especially in resource-limited settings and those missing adequate structures with health care professionals [18,19]. Furthermore, the demographic development and the increase in complexity due to the long duration of HIV disease require integrated strategies for patient care [20]. The HIV-specific multifactorial situation combined with smoking [21], drug abuse anxiety, depression, and addictive disorders proves that PLWH need an adequate care strategy in a setting of health care professionals from multiprofessional disciplines [22,23]. Moreover, PLWH do not seek appropriate specialists when needed due to the fear of discrimination and stigmatization [24]. Furthermore, in the context of reproductive medicine, especially for women with HIV, integrated care models for women of childbearing potential are beneficial, both during pregnancy and after birth, from the perspective of the mother and the newborn [25,26]. The diagnosis and treatment of accompanying diseases (tuberculosis and hepatis B and C, especially in African countries) cannot be treated optimally in non-specialized treatment centers [19,27].

The effectiveness of ART to suppress the viral load dramatically reduces the likelihood of onward transmission, and treatment as prevention is becoming increasingly evident [28,29,30]. Using a wide variety of additional diagnoses of PLWH as examples, the data also demonstrate that integrated care reduces hospitalization in the clinical setting and saves high costs faced by health care systems [31,32]. Throughout different health care systems, integrated care is a viable solution to the growing demand for improved experiences and health outcomes of patients with multimorbidity and those requiring long-term care [33].

IC guarantees the discharge of the health care system through an interaction of services from different professional groups [34,35]. Health policy considerations for the care of people with complex chronic diseases and the development of evidence-based guidelines and concepts have become more present [36]. Sustainable approaches to dealing with the complex situation of PLWH with multiple chronic diseases represent the concepts of integrated care but also other disease, case, and care management strategies [37].

## 2. Aims

Integrated care is a comprehensive concept with multiple implementation strategies, and the goal herein was to summarize them and demonstrate their applicability and effectiveness for improved treatment outcomes in PLWHIV.

## 3. Methods

A narrative review with a comprehensive literature search was conducted between March and November 2022. The data extraction of the results was oriented toward the central theme of integrated care regarding PLWH. In contrast to systematic reviews, no standard definition exists for narrative reviews. The data extraction, as well as analysis, was guided by the PRISMA checklist for systematic reviews [38,39].

## 4. Search Strategy

A literature search was performed in the databases Pubmed, CINAHL, and Cochrane. The keywords for the search were HIV/AIDS, integrated care, chronic diseases, disease management, case management, adherence, stigmatizing, discrimination, complication, and cognitive disorder. The search was limited to English- and German-language studies.

## 5. Inclusion and Exclusion Criteria

The target group of the studies was PLWH, possibly in combination with other chronic diseases. Qualitative and quantitative studies, including randomized controlled trials and systematic reviews, were included. Non-empirical studies were explicitly excluded. A total of 119 quantitative and qualitative studies, as well as metanalyses and systematic reviews, were analyzed.

## 6. Results

### 6.1. Challenges in the Care of People with HIV as a Chronic Disease

HIV/AIDS as a chronic disease, despite modern antiretroviral therapy over the course of the disease career, can continue to develop into complex challenges due to the development of comorbidities [40]. Acute medical care related to organic complications of the pre-existing HIV disease is the determining factor. This means that the momentary problem is focused on leaving chronicity in the background, but as a causative condition. Additionally, there is a lack of flexibility and interdisciplinarity, as well as an awareness of an interlocking treatment approach to successfully manage the alternating phases of disease progression between the acute and chronic phases [41]. In addition, the inadequate training, information, and participation of PLWH, as well as of caregivers on the treatment team, lead to avoidable dependence and passivity [42]. Evidence-based guidelines, where treatment pathways and strategies are mapped in a process-based manner and ensure optimal patient-centered care, are either insufficient or do not acquire the attention that they need [35,43]. In particular, by reducing repeated, cost-intensive hospitalizations, a significant reduction in financial and personnel burdens on the health care system, as well as on the patients themselves, can be achieved [44].

### 6.2. Definition of and Strategies for Integrated Care

Today, caring for PLWH with complex health- and disease-related but also psychosocial needs continues to be a major challenge for the health care system to manage. Integrated care appears to be one answer to this problem, with collaboration and integration efforts among components of care systems, professionals, and service providers, aimed at improving the efficiency and appropriateness, as well as patient-centeredness, of care [34,45]. Patients with chronic conditions are challenged to navigate a number of interfaces within the health care system as part of their disease progression. IC addresses the interfaces of the multidisciplinary setting (Figure 1), with the goal of optimizing the transitions between disciplines for diagnostics and therapy, including additional areas, such as psychologists, nurses, social workers, and other counseling professions, in a patient-friendly, holistic, efficient, patient-centered, and effective manner [46].

Through standardized treatment processes of integrated care, both the quality of therapy and economic efficiency are increased by saving costs for health policy [47]. The health care system in many European countries is typified by a variety of different professional groups and specialist disciplines acting in different sub-areas.

This ranges from the activities of general practitioners, special institutes, laboratories, social insurance institutions, and radiology for imaging diagnostics to inpatient and outpatient settings in hospitals, as well as prevention and therapy in rehabilitation centers [48,49]. IC has become established in Europe and represents a form of care management in which the focus is on the treatment process of the chronically ill person [50,51]. There is not yet a standard definition of integrated care. The term arose in the context of managed care in the USA and describes various aspects in the design of patient care, the essential features of which are based on the more optimal networking of different service providers, the mobilization of efficiency reserves, and measures that contribute to quality assurance [52,53]. A significant part of integrated care focuses on the process of treatment rather than the individual case, so the interprofessional and interdisciplinary health care professionals are connected and communicate with each other via interfaces [48].

IC refers to measures and processes that contribute to better interprofessional networking and collaboration among all health care professionals and that optimally coordinate the treatment and care of patients across the entire treatment pathway [54]. On the one hand, integrated care is based on the networking of primary, secondary, and tertiary structures and, on the other hand, on the interdisciplinary coordination of general practitioners, specialists, social workers, case and care managers, nurses, and other health care professionals (physiotherapists and nutritionists) (Figure 1).

The gap of discontinuity between clinical and extramural settings is thus narrowed for patients [55,56]. PLWH in particular benefit from Integrated HIV Care (ICH), a model of care in which specialists from multiple disciplines (Figure 1) collaborate in a clinical setting that is geographically and temporally unrestricted. The goal is to provide PLWH with continuity of care [57,58] in the primary care setting for their complex health problems and the management of ART, co-morbidities, and multimorbidities (hepatitis B and C, intravenous drug use, psychiatric disorders, anxiety, depression, substance abuse, tuberculosis, diabetes mellitus, and hypertension). Integrated care strategies are dependent on health policy funding, organizational and administrative factors, the provision of various needs-based services, and clinical and medical aspects [45].

### 6.3. Concept of Integrated Care

Integrated care has replaced old forms of care in a needs-based and patient-oriented manner. From a political perspective, the affordability of the health care system is also a motivating factor to implement integrated care.

In a national and international comparison, it was found that various models are available for integrated care and managed care (Table 1). In the GP-relief model, the aim is to offer patients with multimorbidities needs-oriented care, with the emphasis on teaching everyday skills to achieve a better quality of life and to have a positive influence on the course of the disease (source). In an international comparison, it was found that, in Spain, there is the Badalona Serveis Assistencials (BSA) program, which is focused on the care of older people with frailty and complex and multiple chronic diseases and cognitive impairment. It is a person-centered program, with medical services, social support, and emergency care available in a networked fashion [59]. The South Somerset Symphony Program, established in England, is about strengthening primary care, with the Enhanced Primary and Complex Care Model available for complex cases with specially trained health care professionals.

The care programs are available for patients and for family caregivers, where coaching regarding resources is also offered [61].

In Germany, the concept “Casaplus” exists as geriatric case management, where trained case managers coordinate relevant health care services with each other in the context of information, counselling, and support, as well as monitoring, for people with chronic diseases [62].

The Health Network Tennengau is a low-threshold care system between primary care and the hospital, with handover and takeover visits in the inpatient setting, the transmission of doctor’s letters and findings to the general practitioner, a hospice initiative, the dissemination of population health information on weight reduction and dementia, and senior counseling [63]. In addition, the “Diabetes under control” program for optimized diabetes care exists in Austria specifically for patients with diabetes mellitus [64].

### 6.4. Disease Management and Case Management

As defined by the Disease Management Association of America [65] disease management is a multidisciplinary, continuous approach to health care for populations with defined conditions or at risk of developing certain conditions. Disease management, or chronic disease management, supports the improvement of the physician–patient relationship; prevents exacerbation and the development of complications and comorbidities through prevention and evidence-based, cost-effective treatment strategies; and involves a continuous evaluation process of medical, economic, and patient-centered outcomes.

Accordingly, disease management means the degree to which patients, health care providers, and other members of the health care professional community follow a rational therapeutic strategy [66].

In contrast to disease management and integrated care (Figure 2), case management is another important organizational form in integrated care and is focused on a specific patient group (those who are seriously ill, those who are disabled, and the elderly). It is a targeted case-based care strategy. Case management can therefore also be described in the broadest sense as an individualized disease management strategy. In case management, certain patients (e.g., patients after organ transplantation, patients with AIDS, patients after stroke or craniocerebral trauma, and oncology patients with complex antitumor therapies) are identified within the framework of care, and they are characterized by peculiarities in their care history due to frequent (re)hospitalizations, repeated or even failed surgical interventions, complicated medical courses of disease and treatment, and complex therapy regimens. This group of patients usually require medical, nursing, psychological, social, rehabilitative, and psychiatric care. Patients are usually not able to cope with this complexity of requirements on their own. This multidisciplinary task is taken over by case managers, who provide a concrete coordination of the care process, including advice on social or financial matters [67]. Additionally, in the interdisciplinary care context for PLWH, service providers such as “buddy programs,” peer workers, voluntary and certified counselors are also involved. Self-help organizations for the informal exchange of experiential knowledge in coping with HIV are also particularly helpful for both patients and family members.

### 6.5. The Complexity of HIV

Complex chronic diseases are characterized by medical, psychosocial, socioeconomic, and psychiatric determinants (Figure 3).

The complexity of HIV (Figure 4) has shifted due to the potential for rapid diagnosis and immediate treatment [70]. Although ART has improved in terms of dosing frequency and low side effects, adherence to the treatment regimen remains a challenge for many PLWH and is the main reason for ART failure [71]. If viral replication suppression is inadequate, the risk of developing resistance to ART agents increases [72]. Furthermore, complications and AIDS-defining sequelae, carcinomas, and infections arise from the reduction in the CD4 cell count and, thus, the destruction of the immune system [73]. The prevention of transmission is also no longer ensured. Poor adherence to ART is related to the complexity of the treatment regimen, medication side effects, patient beliefs about treatment efficacy, and access to care [74]. In addition, complexity arises from HIV itself (Figure 4), as well as from the consequences of the long-term toxicity of ART, comorbidities, substance abuse, alcohol abuse, mental disorders such as depression, hepatitis C infections, type 2 diabetes mellitus, heart diseases, obesity, metabolic disorders, hypertension, and kidney diseases [75,76]. To effectively prevent comorbidities, ICH involves specialists of internal medicine, psychiatry and ophthalmology and individuals from other professions, such as social workers, in primary HIV care at an early stage [70,72].

The specificity of presenting the individual components of the different categories comes from the clustering of the individual aspects and the exacerbation of the overall situation. The challenge in the care of PLWH often lies in the management of social factors [72]. HIV management includes medication and care optimization, symptom management, self-care education, comorbidity management, and continuity care, as well as emotional and social support. For the treatment of common depression, collaboration between different disciplines also delivers an improvement in PLWH in the primary care setting as a disease management program [77].

### 6.6. Benefits of Integrated Care for PLWH

The benefits of integrated care for PLWH have been researched and mapped thematically in the current literature in a very heterogeneous way. The greatest benefit of integrated care is the saving of financial resources. Significant benefits from the patient’s perspective are reduced waiting times and treatment times and less frequent contact with physicians due to the optimized coordination of individual health care professionals. In addition, duplicate examinations and findings are avoided. Furthermore, transparency and the flow of information are increased, and a structured treatment plan based on objective criteria is used. Structured care also enables quality control and assurance for chronically ill patients.

The benefits of integrated care include enormous potential savings, especially through the improved management of interfaces, whilst maintaining and improving the quality of treatment. The avoidance of unnecessary additional examinations, shorter sick leave, and the efficient and effective provision of medicines and other remedies free up resources that can be used for other important areas. Integrated care benefits not only patients, taxpayers, and health policy makers but also the service providers themselves. They follow proven, evidence-based treatment pathways, as well as guidelines, thus optimizing treatment quality and patient outcomes. In addition, the successful treatment of patients without hurdles and waiting times increases overall satisfaction and enhances planning reliability, as well as patients’ motivation and consistency to adhere to and cooperate with treatment management [78]. The benefits of integrated care are primarily argued by the continuous, process-driven, planned interprofessional and multidisciplinary collaboration of individuals from the most important health care disciplines, especially in the extramural setting.

People living with HIV are considered a particularly vulnerable patient population due to their complex medical and psychosocial needs.

The provision of a comprehensive multidisciplinary care system has been the most effective option for PLWH for effective and continuous treatment. Unlike countries with a high HIV prevalence, a large population density, poverty, a lack of hygiene standards, no full access to health care or access to ART, high-income countries additionally face stigma and discrimination [79]. Due to the continuum of complications, additional symptoms with burdens, polypharmacy, and the resulting side effects and interactions, HIV/AIDS can occur in a very complex condition image [72]. Resilience in particular is also necessary for PLWH to cope, and a complex disease course can result because PLWH with low resilience become particularly vulnerable [80]. This vulnerability is seen as post-traumatic stress, depression and anxiety, and a lack of adherence. The concept of integrated care can identify and treat mental health symptoms, such as anxiety and adherence issues, as well as other effects, in a timely manner through timely assessments [81]. The Kaiser Permanente (KP) model for chronically ill patients from the United States is suitable for adapting integrated care to PLWH. The KP model of IC is based on the stratification of PLWH for the targeted, needs-based delivery of various services. In addition to the treatment of the HIV infection, the services include prevention and monitoring and the minimizing of risk factors for potential complications and resulting comorbidities. PLWH receive additional support in self-managing their disease, with the goal of achieving the highest possible level of health literacy [82]. PLWH as high-risk patients receive disease and case management combined with targeted professional care from health care professionals. The core components of the KP model are prevention, self-management support, disease management, and case and care management (Figure 5). The first component of integrated care is based on HIV prevention for the entire population. PLWH are included, provided that patients have a high adherence with the outcome: Undetectable is Untransmittable (U=U) [1,83]. The next stage is characterized by support for the development of self-management strategies for PLWH as chronically ill patients with stable situations. If needed, health care professionals in integrated care are available to PLWH. If the patients have a high level of health literacy, PLWH make targeted and independent decisions to use the appropriate service. The penultimate stage corresponds to disease management for patients with an already complex disease situation, as well as a high risk of additional complications that have a massive negative impact on the existing HIV infection. The top level is associated with case and care management and the involvement of the entire network of IC [33,84].

## 7. Discussion

The national and international concepts and models of integrated care, including the benefits for PLWH, could be presented comprehensively in this review. In addition, this review focuses on the complexity of HIV/AIDS in the context of chronic disease. The complexity of HIV infection with the specific problems (ART, adherence, disease management, comorbidities, symptoms, side effects, long-term toxicities, lifestyle, obesity, a lack of exercise, nicotine abuse, and alcohol and drug abuse) and implicit experiences and burdens (stigma, discrimination, hopelessness, anxiety, depression, isolation, and loneliness) of PLWH establish the need for integrated care to be a necessary patient-centered approach to treatment [72,85,86]. The term integrated care is not uniformly defined and is often used as a synonym for different models of care [56,63,87]. The collaborative participation of physicians with health care professionals from other professions represent the pillars of integrated care for PLWH [88].

In addition to HIV specialists, PLWH care require psychologists, nurses, and social workers, as well as specialists from other disciplines [89]. Contrary to classical chronic diseases, stigma and discrimination are major determinants that negatively affect disease progression and adherence [86,90]. Through the multiprofessional network of integrated care, PLWH are shown to have stability in their relationships with health care professionals, especially in countries with a lack of infrastructure [91]. Studies clearly demonstrate that a lack of adherence is a trigger for the development of complications and comorbidities [92]. Ferlatte et al. (2022) [93] confirm in their study the associations between poverty, HIV, gender, and racial stigma and the impact of antiretroviral therapy (ART) and viral suppression on adherence. Galani et al., (2021) [94] confirm the complexity of HIV as the early detection, targeted treatment of comorbidities and the prevention of additional diseases through the expansion and improvement of a continuous record of health data to compensate for a lack of interprofessional communication. Additionally, this has been shown to prevent the majority of comorbidities (tuberculosis, hepatitis C, the burden of fatigue, and cancer) [95,96]. Due to the multiplicity of symptoms, there is a mutual clustering and, thus, the amplification of individual symptoms (depression, pain, fatigue, anxiety, loneliness, and isolation), with the potentiation of the overall burden and an increase in costs for the health system quality of life [47,97,98]. Studies show that particularly vulnerable populations (sex workers, intravenous drug users with and without methadone programs, and homeless people) with HIV do not always benefit optimally from ART without ongoing support from professionals [1,99,100]. Barba et al., (2016) [101] describe the coordination of patient-centered care for PLWH between primary care and the hospital setting, where numerous medical comorbidities with polypharmacy are successfully treated [102]. Further, integrated care has been shown to be particularly important in PLWH in the palliative setting in relation to stigma and poverty, as well as severe untreated pain, with people in the study describing a longing for death due to the pain [103]. The stigma experience starting from health care professionals in clinical settings in countries with high poverty is particularly stressful for PLWH and is largely avoided through integrated care (Islam et al., 2015) [104]. Ferlatte et al., (2022) [93] highlight the link, particularly among MSMs, between experiences of stigma and increased suicidality.

Integrated care counters this through therapeutic impartiality, empathy through compassionate care, social support, and appreciative and non-judgmental communication [105]. Moreover, integrated care aims to promote health literacy with PLWH’s ability to use health services on their own and build resilience and self-efficacy to cope with the complex situation [29,106]. In particular, in remote African countries, life expectancy can be increased by 75 percent compared to people without access to health care [98,107]. Overall, it should be noted that the implementation of all integrated care models and concepts depends primarily on health policy funding [108,109].

## 8. Limitations

The following limitations arose in the context of this review: (1) Due to the heterogeneity and complexity of PLWH as a patient group, children and adolescents were explicitly excluded. (2) Only studies in English and German published in scientific journals were included. This could lead to the exclusion of relevant literature published in gray literature, or studies from other regions using languages such as Portuguese, French, and local African languages. (3) Some studies referred to countries in Africa and India with a high HIV prevalence with limited universal health coverage. The results cannot be generalized to industrialized countries. (4) The chosen methodology does not allow for generally valid global conclusions about the benefits of integrated care. (5) In the studies included, the individual professional groups working together in integrated care sometimes differed significantly. This has implications for the different, country-specific legal competencies of health care professionals. (6) Narrative reviews do not involve rigorous assessments of study quality because the focus is on capturing the range of work that provides information on the topic and not limiting the work to studies that meet certain standards of scientific rigor.

## 9. Conclusions

Our review shows that integrated care for PLWH is potentially feasible and associated with high patient outcomes. In addition, a holistic patient-centered view of PLWH, including medical, nursing, psychosocial, and psychiatric needs, as well as the various interactions among them, must be considered in planning. Moreover, health inequalities for PLWH, such as uniform unrestricted access to health care and interprofessional treatment of comorbidities, as well as AIDS-defining diseases, and for PLWH with substance abuse and social problems need to be implemented in the key competencies of health care professionals. Prospectively, future research on integrated care is particularly needed for PLWH but also for other patients with specific complex diseases.

## Figures and Tables

**Figure 1 ijerph-20-03374-f001:**
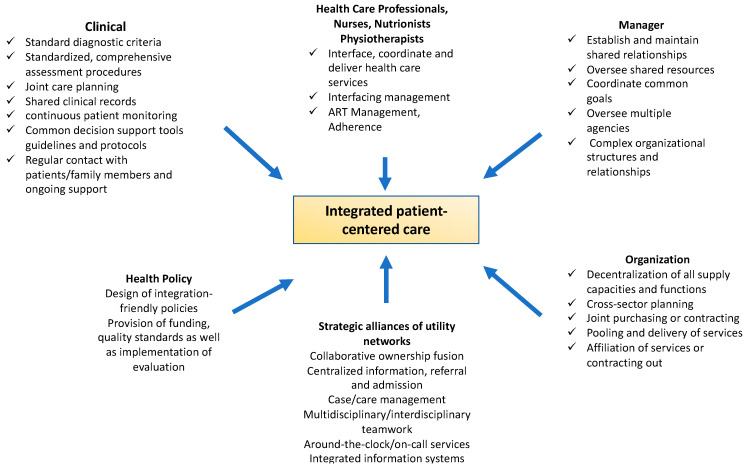
Model of integrated care (own presentation; data from the (WHO) [33].

**Figure 2 ijerph-20-03374-f002:**
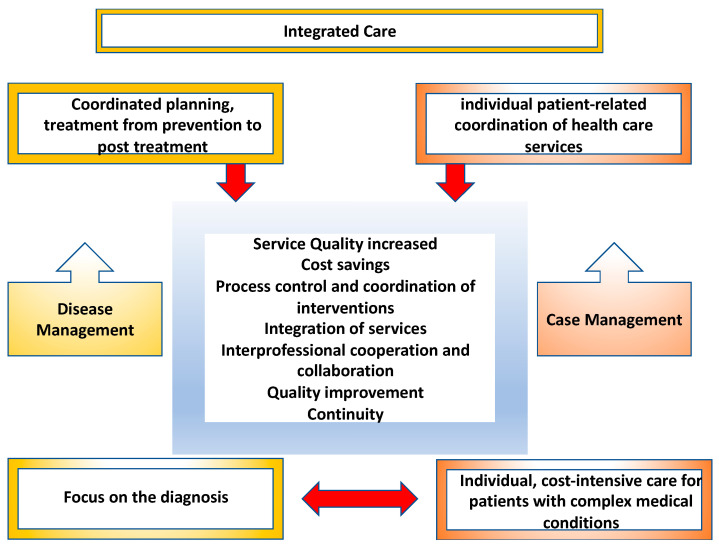
Integrated care, disease management, case management (own presentation; data from Bower, 2014; Taylor, Candy, and Bryar, 2005) [68,69].

**Figure 3 ijerph-20-03374-f003:**
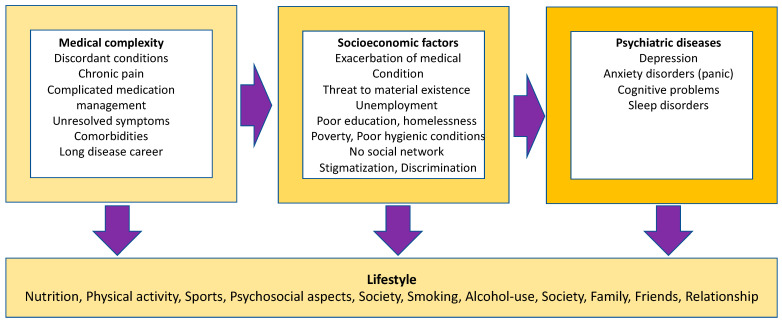
General complexity of people with chronic diseases (own presentation).

**Figure 4 ijerph-20-03374-f004:**
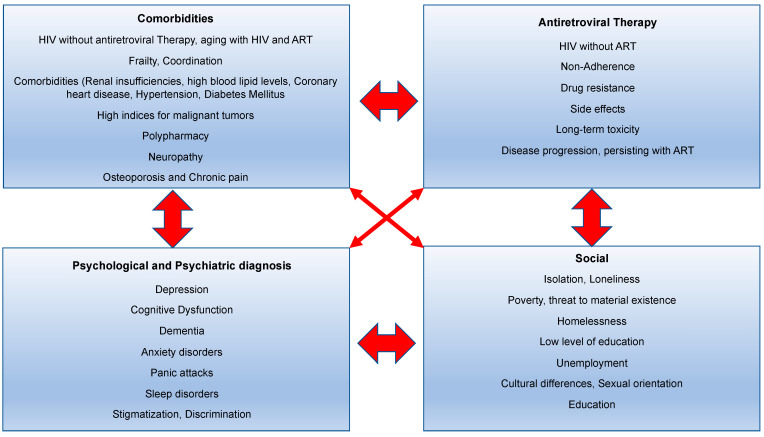
Categories of HIV complexity (own presentation) (Watson, 2018 [70]).

**Figure 5 ijerph-20-03374-f005:**
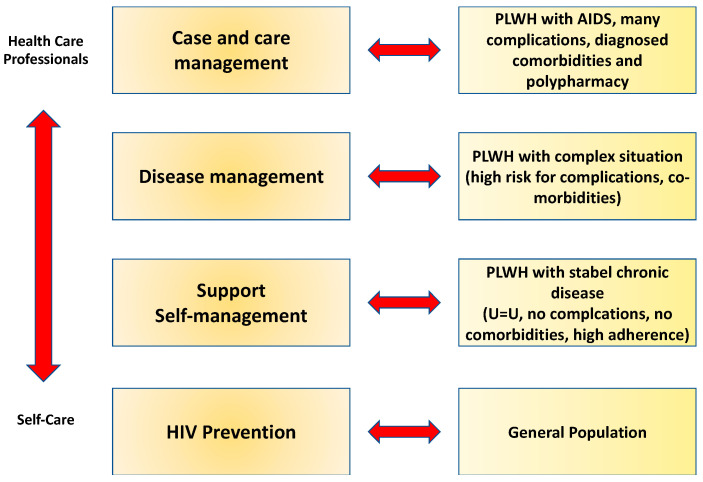
Integrated care for PLWH modified population-based Kaiser Permanente model (WHO 2016 [33]).

**Table 1 ijerph-20-03374-t001:** Overview of national and international integrated care programs (own presentation; data from Institute for Advanced Studies Austria 2016 [60]).

Austria	Health Network Tennengau Sociomedical Center Liebenau Disease management programme “Therapie aktiv—Diabetes im Griff“
Croatia	Geros Palliative Care System
Germany	Casaplus Gesundes Kinzitgal
Hungary	Onko Network Palliative Care Consult Service
The Netherlands	Proactive Primary Care Approach for Frail Elderly Care Chain Frail Elderly Better Together in Amsterdam North
Norway	Medically Assisted Rehabilitation Bergen Learning networks for whole, coordinated, and safe pathways
Spain	Badalona Serveis Assitencials Area Integral de Salut, Barcelona Esquerra
United Kingdom	Salford Integrated Care Programme/Salford Together South Somerset Symphony Programme

## Data Availability

Data is not available due to privacy and ethical guidelines.

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
