# Peer review of "Integrated Care as a Model for Interprofessional Disease Management and the Benefits for People Living with HIV/AIDS"

_ijerph, 2023, doi:10.3390/ijerph20043374_

Round 1
Reviewer 1 Report
The paper addresses the important topic of organising coordinated health care for patients living with HIV (PLWHIV). No model of coordinated care that can be considered exemplary and fully responsive to patients' needs is in place in the current healthcare systems in Europe; hence the reviewer considers the choice of the subject of the study to be important.
The study's objective was correctly identified, the methodological approach was correct, the PRISMA procedure was applied, and the inclusion and exclusion criteria adopted are sufficient in the reviewer's opinion. The study's introduction is appropriate and corresponds to the current literature on the subject (which is extremely important because, due to the intensive development of work on HIV policy, publications from many years ago may already be outdated).
In the reviewer's opinion, the presented model solutions for PLWHIV care lack (which may be due to the lack of such solutions in the analysed literature) the influence of NGOs, voluntary and professional sexual health clinics run by NGOs, as well as non-medical certified HIV/AIDS counsellors. Such counsellors exist in the systems of some countries (e.g. Poland, USA); their support does not only consist in testing healthy people but also in counselling PLWHIV, mainly in the area of acceptance of their infection. There are also appearing, co-financed from European funds (Norwegian and Swiss funds), programmes of support for people with experience of HIV infection ("buddy programs", peer-workers, peek-assistants), which provide systemic support for patients who have found out about their infection and are in the process of accepting this situation.
In the presented models of care (e.g. in the diagrams), the authors point out the interdisciplinary nature of patient care and recommend including nursing care, which is extremely important in the reviewer's opinion. Still, I recommend supplementing the proposal with support from NGOs, peer workers, and certified counsellors. Such an arrangement could help to strengthen the coordination and comprehensiveness of care.
In the paper, the authors do not build a recommended model of coordinated care; however, pointing to its benefits (chapter 6.6.), which, in the opinion of the reviewer, is the right move, as it is difficult to propose solutions common for such dispersed and diverse systems operating in different countries.
For the rest, the reviewer sees no need to modify the work.
Reviewer 2 Report
In its entirety, I found the introduction rather difficult to understand and read through. There are instances where the emphasis of the introduction is clear, but there is no clear logical flow. This needs addressed, with a logical flow that relates to the rest of the paper (i.e., aims/methods). It is significantly missing conciseness and clarity.
The overall aim of this paper is somewhat distinguishable, particularly once the authors arrive to the results section. However, throughout the introduction, in several parts throughout the results section, as well as the discussion section, there is a significant lack of clarity on what the paper is referring to. As the authors mention, there is no clear definition of what integrated care really is. The authors address this within the context of HIV/AIDs; however, the overall quality, explanation, and description of the relevance of this topic is not made clear to the reader.
Round 2
Reviewer 2 Report
No further comment.